Polylactic acid as a suitable material for 3D printing of protective masks in times of COVID-19 pandemic

http://orcid.org/0000-0001-8863-0696 Vaňková Eva 1 Eva.Vankova@vscht.cz
http://orcid.org/0000-0003-2182-0430 Kašparová Petra 1
Khun Josef 1
Machková Anna 1
http://orcid.org/0000-0003-3780-2966 Julák Jaroslav 1 2
Sláma Michal 3
Hodek Jan 4
Ulrychová Lucie 4
Weber Jan 4
Obrová Klára 5
Kosulin Karin 5
Lion Thomas 5 6
Scholtz Vladimír 1
1 Department of Physics and Measurements, University of Chemistry and Technology Prague , Prague , Czech Republic
2 Institute of Immunology and Microbiology, First Faculty of Medicine, Charles University and General University Hospital , Prague , Czech Republic
3 Faculty of Science, University of Hradec Kralove , Hradec Králové , Czech Republic
4 Institute of Organic Chemistry and Biochemistry of the Czech Academy of Sciences , Prague , Czech Republic
5 Children’s Cancer Research Institute , Vienna , Austria
6 Department of Pediatrics, Medical University of Vienna , Vienna , Austria
Joshi Sanket
Electronic publication date: 2020 Oct 29
Publication date: 2020
Volume: 8
Electronic Location ID: e10259
Received 2020 Jun 19; Accepted 2020 Oct 6
Copyright: © 2020 Vaňková et al.
Copyright year: 2020
Copyright holder: Vaňková et al.
License: This is an open access article distributed under the terms of the Creative Commons Attribution License, which permits unrestricted use, distribution, reproduction and adaptation in any medium and for any purpose provided that it is properly attributed. For attribution, the original author(s), title, publication source (PeerJ) and either DOI or URL of the article must be cited.
License URL: https://creativecommons.org/licenses/by/4.0/

Keywords: COVID-19, Disinfection, Polylactic acid, Protective masks, SARS-CoV-2, 3D printing, Reusable material, Virological testing, Ethanol, Human adenovirus

Funding: The authors received no funding for this work.

==============================
A critical lack of personal protective equipment has occurred during the COVID-19 pandemic. Polylactic acid (PLA), a polyester made from renewable natural resources, can be exploited for 3D printing of protective face masks using the Fused Deposition Modelling technique. Since the possible high porosity of this material raised questions regarding its suitability for protection against viruses, we have investigated its microstructure using scanning electron microscopy and aerosol generator and photometer certified as the test system according to the standards EN 143 and EN 149. Moreover, the efficiency of decontaminating PLA surfaces by conventional chemical disinfectants including 96% ethanol, 70% isopropanol, and a commercial disinfectant containing 0.85% sodium hypochlorite has been determined. We confirmed that the structure of PLA protective masks is compact and can be considered a sufficient barrier protection against particles of a size corresponding to microorganisms including viruses. Complete decontamination of PLA surfaces from externally applied Staphylococcus epidermidis, Escherichia coli, Candida albicans and SARS-CoV-2 was achieved using all disinfectants tested, and human adenovirus was completely inactivated by sodium hypochlorite-containing disinfectant. Natural contamination of PLA masks worn by test persons was decontaminated easily and efficiently by ethanol. No disinfectant caused major changes to the PLA surface properties, and the pore size did not change despite severe mechanical damage of the surface. Therefore, PLA may be regarded as a suitable material for 3D printing of protective masks during the current or future pandemic crises.

Introduction

COVID-19 (coronavirus disease 2019) is the designation of the disease caused by the SARS-CoV-2 infection. The World Health Organization (WHO) declared this epidemic a global pandemic affecting the whole world on 11 March 2020.

The infection by SARS-CoV-2 was confirmed for the first time in Wuhan, China, but had a huge impact also in Europe and later in North and South America. Lombardy, Italy was the most severely affected region in Europe. Due to the risk of health care system collapse, the Italian government ordered a nationwide lockdown (Spinelli & Pellino, 2020). Several studies showed that SARS-CoV-2, similarly to SARS-CoV-1, remains infectious for hours and days in aerosols and on surfaces, respectively (Chin et al., 2020; Kampf et al., 2020; Van Doremalen et al., 2020), emphasizing the need for efficient virucidal disinfection. The number of patients suffering from COVID-19 disease and the enormous rate of infection spread caused serious complications in many countries, including a desperate lack of protective equipment (Swennen, Pottel & Haers, 2020). Sufficient production and distribution of protective equipment has been crucial for sustaining patient care during the pandemic. The current unsatisfactory situation regarding protective equipment in the USA has been described by Ranney, Griffeth & Jha (2020).

Because of the lack of protective equipment including face masks, extended manufacturing facilities have become very important for supporting the health care system. In this regard, the production of protective masks using 3D printing has proven very promising. This technology, often based on Fused Deposition Modelling (FDM) due to its cost and technical benefits, has found various applications in the manufacturing of medical devices such as prosthetic and dental implants or scaffolds in tissue engineering (Roopavath & Kalaskar, 2017; Tack et al., 2016). The properties of 3D-printed objects render this technology attractive for manufacturing of protective masks. FDM provides adequate dimensional control, good surface finish and adaptability to use a variety of thermoplastic polymer filaments. The technology is based on high-temperature sintering of filaments and subsequent solidification of the printed product at room temperature. The polymers most commonly used for FDM are acrylonitrile-butadiene-styrene copolymers, polycarbonate, polyethylene terephthalate glycol (PETG) and polylactic acid (PLA) (Chadha et al., 2019; Ngo et al., 2018). Due to its unique properties, PLA is one of the most attractive materials for 3D printing. Its main advantages include low printing temperatures of 200–210 °C, smooth appearance, low toxicity and favorable mechanical properties, especially a low warping effect and high geometric resolution (Pajarito et al., 2019; Vicente et al., 2019).

PLA is a biodegradable linear aliphatic polyester produced from renewable natural resources such as corn, wheat or sweet sorghum (Nampoothiri, Nair & John, 2010). Nagarajan, Mohanty & Misra (2016) comprehensively reviewed its properties and applications. This polymer is produced by acid-catalyzed polycondensation of lactic acid monomers. Lactic acid of any chirality can be used, resulting in either poly-L-lactic acid, poly-D-lactic acid or poly-L,D-lactic acid (consisting of both isomers). Since L-lactic acid is the most common isomer in nature and is easily produced by lactic fermentation of various bio-wastes by bacteria (e.g., Lactobacillus spp.), it is also the most commonly used precursor for PLA manufacturing. The possibility of biotechnological production of the monomer significantly decreases its price, making the production of PLA very cheap. The glass transition temperature of PLA ranges between 50 and 80 °C, and the melting temperature reaches approximately 175 °C. Due to its natural precursor, PLA is easily biodegradable, for example, by thermal decomposition, enzymatic digestion, oxidation or photolysis. Ghorpade, Gennadios & Hanna (2001) studied the outcome of PLA-composting for 90 days and found that the compound was degraded by 70 %. The use of PLA is limited by its poor thermal stability and easy hydrolysis—it degrades more easily than other aliphatic polyesters. Nevertheless, PLA has found many applications in diverse areas including the packaging industry as a food packaging polymer for short shelf life products, the pharmaceutical industry for controlled drug delivery formulations and for tissue regeneration, and agriculture for better herbicide delivery management without negative effects on crop yield (Auras, Harte & Selke, 2004; Aziz, Haq & Raina, 2020; Farto-Vaamonde et al., 2019).

Protective masks made by 3D printing from PLA are designed for repeated use, requiring frequent cleaning and disinfection. The low glass transition temperature and relatively low melting point of PLA makes heat sterilization in an autoclave at 121 °C impossible (McKeen, 2014). The polymer can be sterilized using ethylene oxide, gamma radiation (Fleischer et al., 2020) or dry heat below 80 °C for no more than 20 min (Zou et al., 2011). Fleischer et al. (2020) examined the changes of PLA properties after cleaning with chemical disinfectants such as Cidex Opa (Johnson & Johnson) or chlorine solutions. Although these substances caused minor changes in stiffness and strength of 3D-printed PLA, 3D printing at appropriate conditions makes PLA objects mechanically amenable to cleaning and reuse. However, surface porosity of 3D-printed PLA medical tools should be minimized to prevent exposure of users to residual disinfectants by inhalation or skin contact. Oth et al. (2019) studied PLA object sterilization by low-temperature hydrogen peroxide gas plasma in the commercially available Sterrad® apparatus (Johnson & Johnson). They observed only sub-millimeter deformations induced by this process, rendering it suitable for sterilization in different areas including surgical applications. In contrast to conventional steam autoclaving, sterilization by hydrogen peroxide prevents deformation of 3D-printed objects made from PLA or PETG. Swennen, Pottel & Haers (2020) presented a prototype of reusable custom-made 3D-printed face masks (produced by a selective laser sintering technique) from polyamide composite components. The authors proposed cleaning by 15 min exposure to a broad-spectrum antimicrobial solution, ANIOS CLEAN EXCEL, containing didecyldimethylammonium chloride and chlorhexidine digluconate. Nevertheless, material leakage and virus decontamination of the reusable face mask components have not been tested upon one or more disinfection cycles.

In the present study, we have investigated FDM 3D-printed PLA structure and porosity after exposure to common chemical disinfectants including ethanol, isopropanol and a commercial disinfectant containing sodium hypochlorite, which are easily accessible. In addition, we examined the efficiency of PLA disinfection after artificial contamination with bacteria (Staphylococcus epidermidis, Escherichia coli), a yeast fungus (Candida albicans), viruses (SARS-CoV-2 and human adenovirus – HAdV) or natural contamination by wearing the masks.

Materials and Methods

PLA material and masks preparation by 3D printing

Polylactic acid (PLA) was purchased in the form of filament for FDM 3D printing from Shenzen Creality 3D Technology Co., LTD, China. Protective masks, circular plates (diameter of 10 cm and height of 0.2 cm, printed vertically) and square carriers (1 × 1 cm, 0.2 cm high) (Fig. 1) were prepared using a 3D printer (Prusa i3 MK3, Czech Republic). The printing template was designed with Trimble Sketchup Pro, exported in a stereolithographic (stl) file (freely available at https://www.facebook.com/groups/1346383268879783/files/) and used to print the objects of investigation. The printing parameters were as follows: layer height = 0.3 mm, shell thickness (perimeter) = 0.4 mm, bottom/top thickness = 0.2 mm, fill density = 10%, print speed = 90 mm/s, extrusion temperature = 215 °C, platform temperature = 60 °C, filament flow = 95%, machine nozzle size 0.4 mm, the infill pattern was grid (i.e., linear tilted 45°) and the total layers = 338.

Figure 1 Objects made from PLA filaments using 3D printing by the FDM technology.

(A) PLA carriers (1 × 1 cm). (B) Circular plate with a diameter of 10 cm (printed vertically). (C–E) Different types of PLA masks.

Visualization of PLA mask structure using scanning electron microscopy

The structure and porosity of PLA 3D-printed masks were examined using a scanning electron microscope (SEM) Nova NanoSEM 450 (Fei, USA). Approximately 1 × 1 cm pieces cut from printed masks were completely air-dried and visualized by SEM. Since the material is very sensitive to electron exposure, mild conditions had to be used, that is, voltage of 5 kV and low vacuum. Images of each visualized position were captured by LVD detector at gradual magnifications 2,000×, 1,000×, 500×, 100× (focusing on identical position), dwell time 5 µs and spot size 4.5. The size of the pores between PLA filaments was measured and marked using a SEM operating software (xT microscope Control v6.3.4 build 3233), provided by the SEM manufacturer (Fei, USA). The SEM images shown in this study were selected as representative visualizations of the PLA microstructure. The SEM analysis merely illustrates the 3D-printed PLA surface morphology, and gap width measurements are not analyzed statistically.

Visualization of PLA mask structure under stress conditions using scanning electron microscopy

To investigate the impact of possible stress factors for PLA masks, cleaning with chemicals was performed and exposure to wearing-associated contamination was simulated, as outlined below.

The effect of immersing in three chemical disinfectants (96% ethanol, 70% isopropanol and the commercial disinfectant and bleach SAVO Original, Unilever ČR s.r.o., Czech Republic containing 0.85% sodium hypochlorite diluted with water (2:9)) was tested by repeated (5 × 15 min) cycles and long-term (24 h) exposure.

The simulation of human impact on the PLA structure was performed as follows: extensive exposure to fingers (to simulate incorrect application of the mask), abrasion with paper (minor mechanical stress) and dining fork (strong mechanical stress), immersion in 1.9% sodium chloride solution for 4 h (to simulate perspiration).

Short rinsing with 100% acetone was examined to investigate its effect on PLA surface properties. After each treatment, completely dried PLA carriers were examined using SEM, as described above.

Aerosol particle passage through PLA material

Surface contamination (and potential surface penetration) with infectious agents was simulated by an aerosol generator and a photometer (Lorenz Meβgerätebau FMP 03) with a differential pressure sensor (Fig. S1), providing a suitable test system with stand for facial masks and flat filter materials. The device was certified as a test system according to the standards EN 143 (Respiratory protective devices—Particle filters—Requirements, testing, marking), and EN 149 (Respiratory protective devices—Filtering half masks to protect against particles—Requirements, testing, marking). A sample of the PLA material was attached in the standardized testing cartridge and was sealed with silicone to prevent false positive detection of penetrating particles passing along the edge of the PLA panel (Fig. S2). The cartridge was mounted into the Lorenz Meβgerätebau FMP 03 device, between the aerosol generator and photometer. The aerosol generator produced a defined amount of aerosolized paraffin oil, the test system passed it through the material, and the photometer situated on the other side of the PLA sample measured the aerosol concentration, thereby indicating the retention efficiency. An integrated differential pressure sensor was used to determine the pressure loss during passage through the sample. The particle size distribution was approximately 0.1–2 µm (geometric mean 0.44 µm), which is close to the most frequently observed penetrating particle size (Fig. S3). The output of the aerosol generator was set to 150% with flow 95 L/min, atomizer pressure 5 bar and oil temperature 60 °C. The test was performed for 270 s.

Disinfection of PLA material artificially contaminated with bacteria and yeast fungus

Wild strains of S. epidermidis, E. coli and C. albicans were used as representatives of gram-positive and gram-negative bacteria or yeast fungus, respectively. The concentration of bacteria was adjusted to approximately 1 × 107 colony forming units (CFU) per mL, the fungus concentration was 1 × 106 CFU/mL. Each PLA carrier with a size of 1 × 1 cm was contaminated with 10 µL of microbial suspension applied to the surface of carriers in 1 µL droplets for 1 h. The disinfection of contaminated carriers was carried out by immersing in three mL of 96% ethanol, 70% isopropanol, or 0.85% sodium hypochlorite (SAVO Original, Unilever ČR s.r.o., Czech Republic) for 15 min. After evaporation of disinfectant solutions, the carriers were immersed in one mL of sterile 0.9% saline, vortexed, and the obtained suspensions were inoculated onto appropriate agar plates. Blood agar was used for S. epidermidis, Müller-Hinton (Oxoid, Czech Republic) agar for E. coli and Sabouraud agar (Oxoid, Czech Republic) for C. albicans. Samples not exposed to treatment by disinfectants were used as controls. The inoculated plates were incubated at 37 °C for 48 h. Each experiment was done in triplicate, and results were obtained by counting the average CFU/mL.

Disinfection of PLA material artificially contaminated with viruses

SARS-CoV-2, the causative agent of the COVID-19 pandemic, was isolated in a biosafety level 3 laboratory from a nasopharyngeal swab by inoculating Vero CCL81 cells (ECACC 84113001) and subsequent expansion by two additional passages in Vero CCL81 cells. Passage 3 was cleared by centrifugation at 1000 g for 5 min, passed through a 0.45 µm filter, and stored at −80 °C until use. In addition to SARS-CoV-2, inactivation of a stable DNA virus, the Human Adenovirus 2 ATCC VR-846 (HAdV) obtained from the American Type Culture Collection (ATCC) was assessed.

Similar to the previous set of experiments, PLA carriers of 1 × 1 cm size were contaminated with 20 µL of a SARS-CoV-2 suspension displaying a median tissue culture infectious dose (TCID50) of 106 IU/mL, which was applied to the surface of carriers in 1 µL droplets. An additional set of carriers was covered with 50 µL of HAdV suspension (106 virus copies) spread evenly over the entire surface. The contaminated carriers were then immersed in 96% ethanol, 70% isopropanol or 0.85% sodium hypochlorite for 15 min. Subsequently, residual viruses—if present—were washed from the dried surface using 180–200 µL PBS. The solution was used directly for infection of Vero-E6 cells (ATCC CRL-1586), in case of SARS-CoV-2, or A-549 human lung carcinoma cells (DSMZ ACC107 from German Collection of Microorganisms and Cell Cultures), in case of HAdV, respectively. Recovered SARS-CoV-2 was titrated by an immunofluorescence (IF) assay using a 1:2.5 serial dilution of Vero-E6 cells starting from 10 µL. Vero-E6 cells were incubated for 72 h at 37 °C in a CO2 incubator prior to the IF assay. Briefly, medium was washed out, cells were fixed using 4% paraformaldehyde (PFA), cell membranes were perforated with 0.2% Triton-X100, and SARS-CoV-2 was labeled with primary mouse anti-SARS-CoV-2 antibody. Secondary anti-mouse antibody was conjugated with a Cy3 fluorophore and a fluorescent microscope (Olympus IX 81, Germany) was used for signal detection. In the case of HAdV, serial dilutions of virus inoculum were used to infect A-549 cells and the cytopathic effect (CPE) was determined using Motic AE21 Inverted Phase Contrast Microscope (Zeiss, Germany). The titers of both recovered viruses infection particles were determined as TCID50 and calculated using the Spearman-Kärber method (Kärber, 1931; Spearman, 1908). In addition, recovered HAdV genome copies were determined by real-time quantitative PCR (RQ PCR) as described previously (Lion et al., 2003) using the ABI Prism Fast 7500 Instrument (Thermo Fisher Scientific, MA, USA).

Disinfection of PLA masks worn by test persons

To investigate the feasibility of disinfecting PLA protective masks in practical use, three volunteers wore the protective masks of the same type for 4 h. Thereafter, smears from one half of the inner (approximately 80 cm2) or outer surface (approximately 83 cm2) of each mask were performed using sterile cotton swabs. These samples served as a control for natural mask contamination by manual handling, direct skin contact and exhalation. Each cotton swab was transferred into one mL of 0.9% saline in a microtube, vortexed and inoculated onto a blood agar plate. Thereafter, the filters were removed from masks and the PLA skeletons of the masks were immersed in 96% ethanol for 15 min. After ethanol evaporation, cotton swab smears were taken from the second halves of the inner and outer mask surfaces, inoculated onto agar plates, and incubated at 37 °C for 48 h. The results were averaged and expressed as CFU/mL.

Results

Investigation of structure and porosity of 3D-printed PLA material

The structure and porosity of PLA masks produced by 3D printing were investigated by SEM. Scanning electron micrographs of gaps between the PLA filaments were captured at four different magnifications (Fig. 2). The PLA filament size determined was 312.8 µm (Fig. S4) and its surface appeared macroscopically very smooth (Fig. 2A). Further magnification showed only slight roughness of the surface and very small gaps between filaments (Figs. 2B and 2C). Additional increase of magnification revealed connecting filaments of PLA, resulting from the high temperature during 3D printing, with only very small pores (6.049 µm in size) in between. The pores appeared to be completely closed deeper in the carrier, as observed at the highest magnification used (2,000×) (Fig. 2D). To further test whether the pores were indeed closed and prevented particles from passing through the printed mask, we determined the number of paraffin oil aerosol particles displaying a size of 0.1–2 µm using the aerosol generator and photometer, certified as a test system according to the common standards. Maximum pressure loss of the generated aerosol was detected, and absolutely no penetration occurred even though the PLA sample was printed with a diameter of 10 cm (corresponds approximately to the printed height of the masks) in the vertical position, simulated printing at a lower temperature in the upper layers (on the z-axis).

Figure 2 Scanning electron micrographs depicting the structure and porosity of PLA material used for 3D printing of protective masks.

(A) Magnification 100×, scale bar 500 µm. (B) Magnification 500×, scale bar 100 µm. (C) Magnification 1000×, scale bar 50 µm. (D) Magnification 2000×, scale bar 30 µm. The observed gaps were measured and marked by black lines. SEM parameters: low vacuum, 5 kV, LVD detector, dwell time 5 µs, spot size 4.5. Images were taken at various magnifications at the same position.

Effect of ethanol, isopropanol and sodium hypochlorite on disinfection of PLA material contaminated with bacteria, yeast fungus or viruses

The results of disinfection of artificially contaminated PLA are summarized in Tables 1 and 2. Although the untreated PLA carriers were contaminated by highly concentrated bacterial suspensions of 1 × 105 CFU/mL, complete decontamination by all disinfectants used was achieved. Single colonies were observed in the samples of S. epidermidis and E. coli disinfected by isopropanol, but these isolated findings can reasonably be considered a contamination that occurred after treatment of the samples. The disinfection of PLA carriers contaminated with C. albicans (4 × 104 CFU/mL) was complete in all cases.

Table 1 PLA material contaminated by Staphylococcus epidermidis, Escherichia coli and Candida albicans, untreated or treated with ethanol, isopropanol or sodium hypochlorite.

	Untreated	Ethanol	Isopropanol	Sodium hypochlorite	
Colony forming units on PLA material after contamination by bacteria and yeast (CFU/mL)	
S. epidermidis	1 × 105	0	1	0	
E. coli	1 × 105	0	1	0	
C. albicans	4 × 104	0	0	0	
Note:

Results are expressed in CFU/mL, as the mean of triplicate tests. Untreated samples indicate the CFU/mL count present on contaminated carriers.

Table 2 PLA material contaminated by SARS-CoV-2 and HAdV, untreated or treated with ethanol, isopropanol or sodium hypochlorite.

	Untreated	Ethanol	Isopropanol	Sodium hypochlorite	
Virus titers recovered from PLA material after contamination by virus (103 IU/mL)	
SARS-CoV-2	114	0	0	0	
HAdV	338	0.8	4.7	0	
Note:

Results are expressed in 103IU/mL, as the mean of triplicate tests. Untreated samples indicate the virus count on contaminated carriers in IU/mL.

Titers of SARS-CoV-2 and HAdV recovered from disinfected or untreated carriers were determined by IF- and CPE-based assays, respectively. All disinfection agents tested showed complete virucidal effects against SARS-CoV-2. Disinfectants per se exhibited a cytotoxic effects on Vero-E6 cells (Table S1), but this effect was eliminated by serial dilutions during virus titer determination. HAdV infectivity was reduced by ethanol and isopropanol, and completely abolished by sodium hypochlorite. Similar trends were observed by RQ-PCR performed for detecting the HAdV genome copy numbers (Table S2).

Investigation of PLA structure after exposure to ethanol, isopropanol and sodium hypochlorite

The effect of disinfectants on the PLA structure was investigated using SEM. PLA structure, gaps between filaments, and the structure of pores after five 15 min cycles of immersing the carrier in different disinfectants are shown in Fig. 3. Treatment with ethanol (Fig. 3B) resulted in slight melting of the PLA filaments, as compared with untreated PLA (Fig. 3A). The overall PLA structure and surface did not change, but, interestingly, the gap size between the filaments was reduced from the original 6 µm to approximately 850 nm (Fig. 3B). This indicates that ethanol treatment may improve the PLA mask properties with regard to structure density. Similarly, isopropanol treatment did not significantly affect the PLA structure (Fig. 3C). Only slight melting was detectable, resulting in decreased gap size to 3.3–4 µm, in comparison to 6 µm in control samples. Moreover, the surface of filaments remained undamaged. Figure 3D depicts the effect of sodium hypochlorite, which did not alter the surface of filaments, but precipitated disinfectant filled the gaps between them, while the gap size remained almost the same as in the control sample (5–7 µm).

Figure 3 Scanning electron micrographs depicting the structure and porosity of PLA material after short repeated treatment (5 times 15 min) with disinfectants.

(A) Untreated sample. (B) Sample treated with 96% ethanol. (C) Sample treated with 70% isopropanol. (D) Sample treated with 0.85% sodium hypochlorite. SEM parameters: low vacuum, 5 kV, LVD detector, dwell time 5 µs, spot size 4.5. Images were taken at 2,000× magnification (scale bar = 30 µm) at the same position. The observed gaps were measured and marked by black lines.

Long-term treatment of PLA by immersion in disinfectants for 24 hours was also investigated using SEM (Fig. 4). The effect of long-term treatment with ethanol (Fig. 4B) was similar to repeated exposure to sodium hypochlorite (Fig. 3D), that is, the gaps between filaments were significantly enlarged to 23.84 µm (Fig. 4B), possibly filled with etched polymer. Investigation of aerosol particle passage through the PLA material after 24 hours in ethanol confirmed that the enlarged gaps were sealed, as no penetration was detected. PLA melting was also observed after prolonged isopropanol treatment (Fig. 4C). The gaps between filaments were sealed with the polymer in an irregular manner, resulting in variable gap sizes ranging from 1.3 to 4.1 µm. As in all previous tests with ethanol, the surface of PLA filaments remained unaffected. In contrast, long-term treatment with sodium hypochlorite damaged the surface of PLA filaments and revealed precipitation of the disinfectant on the surface (Fig. 4D). Similarly to short treatment with sodium hypochlorite, the gaps between filaments, ranging from 2 to 3.5 µm, were completely filled with precipitated sodium hypochlorite (Fig. 3D).

Figure 4 Scanning electron micrographs depicting the structure and porosity of PLA material after long-time treatment (24 hours) with disinfectants.

(A) Untreated sample. (B) Sample treated with 96% ethanol. (C) Sample treated with 70% isopropanol. (D) Sample treated with 0.85% sodium hypochlorite. SEM parameters: low vacuum, 5 kV, LVD detector, dwell time 5 µs, spot size 4.5. Images were taken at 2,000× magnification (scale bar = 30 µm) at the same position. The observed gaps were measured and marked by black lines.

Disinfection of PLA masks by ethanol upon wearing by test persons

To complement the results of disinfection upon artificial contamination (Tables 1 and 2), disinfection of PLA masks after natural use was investigated. The disinfection efficiency with ethanol (96%) is summarized in Table 3. The microbial load detected on the inner surface of untreated masks varied significantly between different users, ranging from hundreds to thousands CFU/mL. Despite this variation, an average of 7 CFU/mL remained detectable after immersing the masks in ethanol for 15 minutes (short rinsing with ethanol was not sufficiently effective; Fig. S5). On the outer surface of untreated masks, 50–150 CFU/mL were detected, and an average of 2 CFU/mL remained detectable after disinfection (Fig. 5).

Table 3 Natural contamination of PLA masks worn by test persons, and subsequently treated with ethanol (bacterial count expressed as CFU/mL).

	Inner surface (untreated)	Inner surface
(after disinfection)	Outer surface (untreated)	Outer surface
(after disinfection)	
Effect of ethanol on disinfection of PLA masks from natural microbial contamination (CFU/mL)	
PLA mask 1	7,000	19	85	0	
PLA mask 2	257	0	153	0	
PLA mask 3	108	2	59	2	
average	2,455	7	99	1	

Figure 5 Effect of ethanol on disinfection of PLA masks contaminated by wearing for 4 h.

A representative set of blood agar plates is displayed. The plates were inoculated with material collected from (A) inner surface of the mask before treatment; (B) inner surface of the mask disinfected by immersion in ethanol for 15 min; (C) outer surface of the mask before treatment; (D) outer surface of the mask disinfected by immersion in ethanol for 15 min.

Visualization of PLA structure upon mechanical and chemical challenge

The impact on the PLA material by finger contact, abrasion by paper or metal and treatment by sodium chloride solution (mimicking perspiration) was analyzed using SEM (Fig. 6). Although fingers may be greasy or sweaty, the contact did not cause any marks or alterations on the PLA surface (Fig. 6A). Similarly, gentle mechanical abrasion with paper did not affect the material (Fig. 6B). By contrast, intensive mechanical scraping with a dining fork significantly damaged the PLA structure (Fig. 6C), leading to compression of PLA filaments, reduction of inter-filament gaps, and shedding of PLA pieces (Fig. 6D). However, neither loosening of filaments, nor increase in gap size or other deformations were observed. Soaking in sodium chloride solution did not affect the structure, but salt crystals were present in the gaps between filaments (Fig. 6E). In addition, the effect of acetone, which is known to damage PLA, was evaluated. Virtually no gap was visible between filaments upon treatment, indicating that even short exposure to acetone smoothens the structure and seals the pores (Fig. 6F).

Figure 6 Scanning electron micrographs depicting the simulation of human impact on structure and porosity of PLA protective masks.

(A) Sample touched by finger. (B) Slightly mechanically stressed sample (paper abrasion). (C) Extremely mechanically stressed sample (scratching by dining fork). (D) Detail of a pore in extremely mechanically stressed sample (dining fork). (E) Sample after immersion in saline solution (perspiration and sweat simulation). (F) Sample after short rinsing with acetone. SEM parameters: low vacuum, 5 kV, LVD detector, magnification 100× or 500×, dwell time 5 µs, spot size 4.5, scale bar 500 or 100 µm.

Discussion

The unexpected and sudden spread of SARS-CoV-2 infection, which resulted in the COVID-19 pandemic, has led to a desperate shortage of personal protective equipment, especially among the frontline workers. Because of this problem, many people started helping each other by manufacturing facial protection equipment from commonly available resources. An intriguing possibility is the production of protective face masks using FDM, the most widespread technique of 3D printing. A variety of polymers are suitable for FDM, including biodegradable PLA as the most affordable and environmentally friendly material because of its natural origin (Ngo et al., 2018). Despite the potential benefits, the suitability of PLA-based materials for protection against viruses was questioned due to their possible high porosity. To the best of our knowledge, this report provides the first data addressing this issue by testing 3D-printed PLA masks (Fig. 1).

The surface and other mechanical properties of products made from PLA or composite filaments were investigated previously (Graupner, Herrmann & Müssig, 2009; Chi et al., 2018; Ivanov et al., 2019; Wang et al., 2016). However, the microstructure of 3D-printed PLA objects is highly dependent on the printing parameters, and it is not possible to predict the structure and porosity of a particular object based on published data. To investigate the surface properties of protective face masks made from PLA, examination of structure and porosity is required. We showed by SEM that 3D-printed PLA masks have a compact structure, with small gaps between filaments. The gaps between individual filaments were 6 µm wide, but higher magnification showed that the pores were not continuous within the PLA carrier (Fig. 2D) and were actually completely closed. This finding was supported by measurements of the filtering efficiency of PLA, which revealed completely blocked passage of nanometer-sized paraffin aerosol particles. The mask material can therefore be considered impermeable for particles displaying the size range tested, including the fungus, bacteria, and viruses investigated. In combination with the obligatory single-use filters complying with FFP2/3 standards, which are inserted into the mask, spreading of the smallest viruses can also be prevented. Moreover, short exposure to acetone resulted in smoothening of the PLA surface (Fig. 6F).

A similar 3D-printed reusable face mask prototype was reported by Swennen, Pottel & Haers (2020). The material (polyamide composite) and the printing method used (selective laser sintering technique) differ from the approach presented, but it provided a proof of principle for 3D printing of individualized 3D face masks with FFP2/3 filter membranes as a feasible and valuable alternative source for protective equipment. However, the authors of the cited study did not perform any virus decontamination testing of the reusable components of the face masks and were hence unable to assess the impact of repeated cycles of disinfection on the properties of the material. It was important therefore to determine the possibility of disinfecting the reusable face mask matrix.

While SARS-CoV-2, being an enveloped RNA virus, belongs to the less challenging pathogens in terms of disinfection, HAdV (non-enveloped DNA virus) is highly resistant to commonly used disinfectants (Gordon et al., 1993; Lion & Wold, 2020). Adenoviruses mostly cause infections with only mild symptoms in immunocompetent hosts (Lion, 2019), but due to their exceptional stability provide a perfect model for testing the inactivation efficiency. In addition, we examined the disinfection of PLA material from contamination with bacteria (S. epidermidis and E. coli) and yeast fungus (C. albicans). These microorganisms are part of the human microbiome and their persistence on the protective mask surface poses a risk for infection and a health threat to mask users (Fisher & Shaffer, 2014). All bacterial and fungal microorganisms studied were successfully disinfected using either 96% ethanol, 70% isopropanol or 0.85% sodium hypochlorite, after immersing contaminated PLA carriers in the respective disinfectant for 15 min (Table 1). Ethanol disinfected the PLA masks contaminated from using by humans (Fig. 5). In comparison to bacteria or fungi, viruses tend to be 1–2 orders of magnitude smaller, making them prone to enter deep into pores of the PLA material. Nevertheless, our data show that efficient disinfection of the PLA carriers from virus contamination is possible, as all tested disinfectants completely inactivated SARS-CoV-2 (Table 2). Treatment with sodium hypochlorite for 15 minutes also completely inactivated the highly resistant HAdV, while ethanol and propanol only led to reduced loads of infectious virus (Table 2). These data are in agreement with the reported sensitivity of both SARS-CoV-2 (Chin et al., 2020; Kampf et al., 2020) and HAdV to specific disinfectants (Gordon et al., 1993; Lion & Wold, 2020). The present findings therefore provide evidence that PLA material disinfection can be performed with comparable efficiency to other surfaces by appropriate exposure to individual disinfectants. The results obtained can conceivably also help design efficient disinfection protocols for protective face masks made from different materials.

Fleischer et al. (2020) examined the changes of PLA material after cleaning with chemical disinfectants (Cidex Opa, Johnson & Johnson and chlorine solutions), revealing mild alterations in the stiffness and strength of 3D-printed PLA samples. However, the authors concluded that high-quality 3D-printed surfaces generated with appropriate printer settings permit cleaning and reuse of 3D-printed medical tools, without compromising their mechanical properties. The authors also stated that immersion in cleaning agents can lead to their absorption into the PLA structure. Thus, additional research is needed to establish efficient and safe chemical cleaning of various 3D-printed surfaces, to prevent health risks associated with tactile and inhalation exposure to chemically cleaned materials.

In general, we observed that five cycles of PLA treatment for 15 min with alcohol-based disinfectants resulted in decreased gap size between PLA filaments, without any remnants of disinfectant visible by SEM. By contrast, sodium hypochlorite precipitate was retained in the PLA structure, filling the gaps between PLA filaments. Disinfection of PLA masks with 0.85% sodium hypochlorite therefore requires further medical investigation to determine whether exposure to the precipitate might be associated with any health risks. Long-term (24-h) treatment of PLA material with disinfectants resulted in partial melting of the filaments, but no erosions of the material were observed (Fig. 4). Ethanol seems to be best suited for the disinfection of PLA masks because it evaporates and does not require removal by rinsing. Moreover, the barrier properties of the mask were not compromised even after long-term exposure, as determined by aerosol challenge.

Although the surface of protective equipment should remain intact, inadvertent contacts with the hands and fingers often occur, and the possibility of inappropriate handling has to be considered. The pandemic setting requires medical staff to wear extensive protective equipment (e.g., overalls, gloves, protective shields and face masks). Such equipment, together with high workload and stress, increases the body temperature and leads to excessive sweating. We mimicked such conditions by mechanical and chemical treatment in order to evaluate alterations of the protective masks. Touching the surface of the PLA material with fingers had no impact, but intensive mechanical stress caused alteration of the PLA filament surface, without affecting the inter-filament gap area. Treatment with sodium chloride (imitating perspiration and sweat) showed salt crystallization in the gaps between filaments (Fig. 6E). Crystallized salt compounds, such as sodium chloride or sodium hypochlorite (Figs. 3D and 4D), can cause discomfort by skin irritation and itching. This issue was described in detail by Payne (2020) and Wollina (2020) who stated that especially front-line workers obliged to wear a single face mask all day suffer from these problems. The exploitation of PLA may solve this issue, because the fast and cheap manufacturing of protective masks made from this material permits production on a large scale, thereby facilitating more frequent mask changes. Additionally, 3D-printed protective PLA masks are biodegradable, with relatively short decomposition time, thereby providing an environmentally friendly solution.

Conclusions

This study shows that PLA material is suitable for protection against various microorganisms as it is not permeable for submicroscopic particles. PLA can be efficiently disinfected from bacteria, yeast fungus, and SARS-CoV-2 by commonly available chemical disinfectants such as ethanol, isopropanol or sodium hypochlorite. However, contamination with HAdV, a highly resistant representative of non-enveloped viruses, could only be completely removed with sodium hypochlorite. PLA material is not altered by the immersion in disinfectant or by manual handling. Possible skin irritation after the use of certain disinfectants needs to be carefully evaluated. Single-use filters meeting the FFP2/3 standards are inserted into the mask structure and will be subject of further research and optimization. Overall, PLA can be recommended as suitable material for the manufacturing of protective face masks at times of epidemic spread of infections, such as the ongoing COVID-19 pandemic.

Supplemental Information

Supplemental Information 1 Testing setup of aerosol generator and photometer Lorenz Meβgerätebau FMP 03 certified as a test system according to the standards EN 143 and EN 149.

The device consists of two parts, main control unit with aerosol generator and laser photometer (on the right of the image) and pneumatically operated filter-holder (on the left of the image).

Click here for additional data file.

Supplemental Information 2 PLA sample with diameter 10 cm printed vertically and placed in standardized cartridge for the aerosol generator and photometer Lorenz Meβgerätebau FMP 03.

On the left of the image is a printed PLA sample. The middle part presents PLA sample attached into standardized Px filter cartridge with a sealer. On the right of the image is presented the whole setup with standardized Px filter cartridge, ready for testing.

Click here for additional data file.

Supplemental Information 3 Aerodynamic particle size distribution in the aerosol generator.

On the x-axis is presented size of generated droplets (μm), on the y-axis is presented a percentage of the total dose. Red line represents geomean of particles, blue line represents a distribution of particles, defined by size and percentage.

Click here for additional data file.

Supplemental Information 4 Untreated PLA filament size.

SEM parameters: low vacuum, 5 kV, LVD detector, dwell time 5 µs, spot size 4.5. Image was taken at 100× magnification (scale bar = 500 µm). The observed gap was measured and marked by yellow line.

Click here for additional data file.

Supplemental Information 5 Effect of ethanol (short rinsing) on disinfection of PLA masks contaminated by wearing for 4 h.

A representative set of blood agar plates is displayed. The plates were inoculated with material collected from (A) inner surface of the mask before treatment; (B) inner surface of the mask disinfected by short rinsing by ethanol; (C) outer surface of the mask before treatment; (D) outer surface of the mask disinfected by short rinsing by ethanol.

Click here for additional data file.

Supplemental Information 6 Cytotoxic effect (CTE) of used disinfectants on VERO E6 cells.

Infectivity of control SARS-CoV-2 virus.

Click here for additional data file.

Supplemental Information 7 PLA material contaminated by HAdV, untreated or treated with ethanol, isopropanol or sodium hypochlorite.

Results are expressed in 103 genome copies/mL, representing the mean of triplicate tests. Untreated samples represent genome copies/mL of carrier without treatment by disinfectant.

Click here for additional data file.

Supplemental Information 8 Raw data for bacteria and yeast fungus contamination.

PLA material contaminated by Staphylococcus epidermidis, Escherichia coli and Candida albicans, untreated or treated with ethanol, isopropanol or sodium hypochlorite. Results are expressed in CFU/mL, as the individual values of triplicate tests. Untreated samples indicate the CFU/mL count present on contaminated carriers.

Click here for additional data file.

Supplemental Information 9 Raw data for SARS CoV-2 contamination.

PLA material contaminated by SARS-CoV-2, untreated or treated with ethanol, isopropanol or sodium hypochlorite. Results are depicted as microimages from brightfield microscopy and immunofluorescence assay.

Click here for additional data file.

Supplemental Information 10 Raw data for adenovirus contamination.

PLA material contaminated by HAdV, untreated or treated with ethanol, isopropanol or sodium hypochlorite. Results are expressed in 103 IU/mL, as the individual values of triplicate tests. Untreated samples indicate the virus count on contaminated carriers in IU/mL.

Click here for additional data file.

Supplemental Information 11 Raw data for SEM images.

Click here for additional data file.

We thank the volunteers from the organization called “3D tiskem proti viru” (eng. version “3D printing against the virus”) as well as Pavel Kubíček, who printed 3D protective masks for people free of charge during the COVID-19 pandemic. We also thank Václav Čeřovský and Jiří Rybáček from IOCB of the CAS and AVEC CHEM s.r.o. for providing the laboratory environment and equipment enabling the project to be conducted.

Additional Information and Declarations

Competing Interests

Author Contributions

Data Availability

The authors declare that they have no competing interests.

Eva Vaňková conceived and designed the experiments, performed the experiments, analyzed the data, prepared figures and/or tables, authored or reviewed drafts of the paper, and approved the final draft.

Petra Kašparová performed the experiments, analyzed the data, prepared figures and/or tables, authored or reviewed drafts of the paper, and approved the final draft.

Josef Khun performed the experiments, analyzed the data, prepared figures and/or tables, authored or reviewed drafts of the paper, and approved the final draft.

Anna Machková analyzed the data, authored or reviewed drafts of the paper, and approved the final draft.

Jaroslav Julák conceived and designed the experiments, authored or reviewed drafts of the paper, and approved the final draft.

Michal Sláma conceived and designed the experiments, performed the experiments, analyzed the data, authored or reviewed drafts of the paper, and approved the final draft.

Jan Hodek conceived and designed the experiments, performed the experiments, analyzed the data, prepared figures and/or tables, authored or reviewed drafts of the paper, and approved the final draft.

Lucie Ulrychová performed the experiments, analyzed the data, authored or reviewed drafts of the paper, and approved the final draft.

Jan Weber conceived and designed the experiments, authored or reviewed drafts of the paper, and approved the final draft.

Klára Obrová conceived and designed the experiments, performed the experiments, analyzed the data, prepared figures and/or tables, authored or reviewed drafts of the paper, and approved the final draft.

Karin Kosulin performed the experiments, analyzed the data, authored or reviewed drafts of the paper, and approved the final draft.

Thomas Lion conceived and designed the experiments, authored or reviewed drafts of the paper, and approved the final draft.

Vladimír Scholtz conceived and designed the experiments, authored or reviewed drafts of the paper, and approved the final draft.

The following information was supplied regarding data availability:

Raw data depicting background microbial experiments and from SEM images are available in the Supplemental Files.

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
