# Peer review of "Polylactic acid as a suitable material for 3D printing of protective masks in times of COVID-19 pandemic"

_PeerJ, doi:10.7717/peerj.10259_

## Round 0.1 · original submission · Major Revisions

PLA for 3D printing masks is quite an interesting proposition for possible applications during this COVID-19 pandemic. It is a well-written manuscript but needs a few clarifications and modifications. Please see the reviewers' comments, who provided some useful insights and suggestions. Kindly consider those comments to further improve your manuscript.

Reviewer 1 ·

Basic reporting

This work evaluates fused deposition modeling 3D printed PLA materials as protective shields against bacteria and viruses. It analyzes material porosity in the context of repeated disinfections and mechanical wear. Tightness is also tested using certified equipment, specifically designed to test the effectiveness of respiratory protective devices.

The paper is clearly written and well structured. The introduction provides the necessary background and explains the motivation in light of the relevant literature. The illustrations are of good quality, but I would eliminate the low-magnification (left) panels of Figs. 3 and 4.

1.1. There is one reporting flaw in this paper. SAVO is a brand name (https://www.unilever.com/news/press-releases/2013/13-04-08-Unilever-Czech-Republic-acquires-SAVO-brand.html), not a product name. Several chlorine-free disinfectants are commercialized under the same brand name (https://www.drogeria-vmd.com/savo-washing-gel-20pcs-white-linen-6278/). Thus, it is important to give a more precise description of the product employed for the disinfection of the PLA samples in this study. To refer to it, please propose an acronym based on chemical composition (e.g. 0.85% NaOCl or NaOCl). The brand name will be mentioned just in the Materials and Methods section, as usual for reagents and instruments.

Experimental design

The methodology is ample and well explained. The work can be replicated based on the text. I would improve the following:

2.1. The term "cytopathic effects" refers to structural changes in cells elicited by viral invasion. If you wish to refer to the impact of a chemical, consider using the term "cytotoxicity". Therefore, on line 265, I would replace "exhibited a cytopathic effects" with "exhibited cytotoxic effects".

2.2. Some raw data is missing:
- Please also present the raw data of filament size measurement (mentioned on line 239).
- Please provide the raw data for the statement "Disinfectants per se exhibited ... (data not shown)..." (line 265).

Validity of the findings

While this work is vast, encompassing several techniques, I have two concerns regarding the validity of the results and the strength of the conclusions:

3.1. Figures 2-4 nicely illustrate the impact of disinfectants on the surface microstructure of the 3D printed samples. Unfortunately, the SEM analysis has not been performed on several spots of each sample to provide descriptive statistics of pore sizes in various conditions (mean+-standard deviation). I would perform such a quantitative image analysis of at least 100 sites per condition and apply the analysis of variance (ANOVA) to test whether the mean gap sizes differ significantly between the 4 groups (untreated, 96% ethanol, 70% isopropanol, and 0.85% NaOCl). In this work, SEM merely highlights certain spots, so quantitative assessments and their interpretation is speculative. For example, it is not clear why long-term exposure to SAVO leads to smaller gaps (Fig. 4D2) than short-term exposure (Fig 3D2). A solid statistical analysis of the change in porosity caused by 5 cycles of 15 minutes of alcohol-based treatments would strengthen the recommendations given in Discussion (lines 387-395) as well as the Conclusions.

3.2. Also, NaOCl precipitates on the mask surface represent a safety concern, acknowledged by the authors (line 391). Since NaOCl is water-soluble, the PLA sample could be washed thoroughly with water after disinfection. The SEM analysis would have been more informative after extensive rinsing. Did it help to remove NaOCl? How are the pores left behind?

Additional comments

PeerJ addresses a broad readership, not just the 3D printing community. Therefore, the Supplementary Material would be much more useful as a single narrative text, written as a complement of the paper's Results section. In this text, please provide a caption for each illustration.

Besides the 4 illustrations currently given as Supplementary Material, I would include at least one more figure, representing the digital models of the 3D printed structures that were tested in the paper. (Porosity and sturdiness do depend on the design.) Such a figure would render the text between lines 250-252 clearer.

·

Basic reporting

This paper investigates polylactic acid as a suitable material for 3D printed personal protective masks.
Overall, the manuscript is well structured and professionally written using clear language. The illustrations used are thoughtfully chosen and appropriately described and discussed. However, several discrepancies appear between the scale bar size depicted in several figures (2B, 2D, 3A2-D2, 4A2-D2) and their description in the corresponding figure captions. I recommend the authors to resolve or explain these discrepancies in the revised version of the manuscript.
In addition, the authors might consider improving the clarity of Figures 2, 3, 4 and 6. For example, by increasing thickness of the yellow segments (used to mark and measure gap width), and/or changing their color, to improve their contrast. Further, the legend containing the SEM parameters could the magnified to increase readability or removed completely (since the relevant information is also included in the figure captions).

Experimental design

I commend the authors for the experimental design of this study. The methodology was appropriately chosen to consider all the practical aspects regarding material decontamination and degradation. In addition, the employed methods are sufficiently explained to enable the replication of their work.

Validity of the findings

The obtained results and their discussion are meaningful. Aware of the limitations of the study (especially related to the retention of sodium hypochlorite precipitate in the masks and its uncertain health risk), the authors proved that PLA is suitable material for the manufacturing of protective masks. Considering the ongoing COVID-19 pandemic and the associated shortage of protective equipment, this study is particularly interesting and useful. I congratulate the authors for the excellent work and recommend their manuscript for publication in PeerJ.

Additional comments

On line 6, there is a typo – the first affiliated institution is marked with letter “a”, while the others are numbered.

Reviewer 3 ·

Basic reporting

How about printing a flexible mask? It could combine PLA with TPU. Check reference:
3D-printed flexible polymer stents for potential applications in inoperable esophageal malignancies.

Experimental design

What is the mechanical property after printing?
The author should provide more details about the structure, like dimensions.
what is the porosity parameter?

Validity of the findings

PLA is a biodegradable material. How does PLA behave under a mask situation?

Additional comments

How does the structure effectively protect?
What is the reaction between PLA and viruses? Will the mask contain the virus? Do you need to clean it after multiple uses?

---

## Round 0.2 · Minor Revisions

A few minor corrections are suggested by reviewers, please incorporate them and submit your revised manuscript.

Reviewer 1 ·

Basic reporting

The manuscript has been improved in comparison to its original version. The text has been revised, illustrations have been redesigned, and the Supplementary Material has been extended. Nevertheless, I still need to express the following concerns:

I further recommended to describe supplementary materials in a single file and provide captions for each illustration. It did not happen. While having a single file would serve the reader's comfort (and, hence, it is optional), figure and table captions are compulsory in scientific writing. The argument that the supplementary materials are commented in the main text of the paper does not hold, since the illustrations included in the paper are also commented and they have a caption, too. For me, Supplementary Figs. 1 and 2 are clear, but I would need a figure caption and English annotations to fully understand Supplementary Fig. 3. The second argument, that supplementary information is unimportant ("not crucial for the confirmation of the claims given in our work and their understanding") is puzzling to me. Why is it supplied then?

I strongly recommend to provide captions for supplementary Figures and for the raw data listed in the following files:
peerj-49258-Raw_data_adenovirus_contamination.docx
peerj-49258-Raw_data_bacteria_and_yeast_fungus_contamination.docx
peerj-49258-Raw_data_SARS_CoV-2_contamination.pdf

Also, please eliminate the brand name "SAVO" from Supplementary Table 1 and from the following raw data files:
peerj-49258-Raw_data_adenovirus_contamination.docx
peerj-49258-Raw_data_bacteria_and_yeast_fungus_contamination.docx

Experimental design

The authors chose not to perform a statistically relevant SEM analysis of inter-filament gap sizes. They consider it too time-consuming. By suggesting it, I did not intend to slow down the publication of this work; I meant to strengthen its message by a thorough analysis of the results. The absence of such an analysis does not invalidate the paper, but it is important to mention, as a limitation of the work, that the SEM analysis merely illustrates the 3D printed PLA surface morphology, and gap width measurements are not analyzed statistically.

Validity of the findings

no comment

Additional comments

I'm sorry that my review is still critical! I tried to respond fast to expedite publication (provided that the other referees recommend it), but I think it is in the interest of all of us (readers and authors) not to leave behind incomplete or confusing text.

·

Basic reporting

no comment

Experimental design

no comment

Validity of the findings

no comment

Additional comments

The authors addressed previous comments well and modified the manuscript accordingly.
For uniformity, the authors might also consider to change the commercial name "SAVO" to "85% sodium hypochlorite" in the supplemental files and raw data provided.

Again, congratulations to the authors for their work!

---

## Round 0.3 · accepted · Accept

Authors amended the manuscript considering all comments and suggestions and the revised version could be accepted in its current form.